# Risk Factors for Delayed Diagnosis of Acute Cholecystitis among Rural Older Patients: A Retrospective Cohort Study

**DOI:** 10.3390/medicina58101392

**Published:** 2022-10-04

**Authors:** Hirotaka Ikeda, Ryuichi Ohta, Chiaki Sano

**Affiliations:** 1Community Care, Unnan City Hospital, 96-1 Iida, Daito-cho, Unnan 699-1221, Japan; 2Department of Community Medicine Management, Faculty of Medicine, Shimane University, 89-1 Enya-cho, Izumo 693-8501, Japan

**Keywords:** acute cholecystitis, delayed diagnosis, abdominal pain, logistic models, aged, rural community

## Abstract

*Background and objectives:* Acute cholecystitis causes acute abdominal pain and may necessitate emergency surgery or intensive antibiotic therapy and percutaneous drainage, depending on the patient’s condition. The symptoms of acute cholecystitis in older patients may be atypical and difficult to diagnose, causing delayed treatment. Clarifying the risk factors for delayed diagnosis among older patients could lead to early diagnosis and treatment of acute cholecystitis. This study aimed to explore the risk factors for delayed diagnosis of acute cholecystitis among rural older patients. *Material and Methods:* This retrospective cohort study included patients aged over 65 years diagnosed with acute cholecystitis at a rural community hospital. The primary outcome was the time from symptom onset to acute cholecystitis diagnosis. We reviewed the electronic medical records of patients with acute cholecystitis and investigated whether they were diagnosed and treated for the condition at the time of symptom onset. *Results:* The average ages of the control and exposure groups were 77.71 years (standard deviation [SD] = 14.62) and 80.13 years (SD = 13.95), respectively. Additionally, 41.7% and 64.1% of the participants in the control and exposure groups, respectively, were men. The logistic regression model revealed that the serum albumin level was significantly related to a time to diagnosis > 3 days (odds ratio = 0.51; 95% confidence interval, 0.28–0.94). *Conclusion:* Low serum albumin levels are related to delayed diagnosis of cholecystitis and male sex. The presence of abdominal pain and a high body mass index (BMI) may be related to early cholecystitis diagnosis. Clinicians should be concerned about the delay in cholecystitis diagnosis in older female patients with poor nutritional conditions, including low serum albumin levels, a low BMI, vague symptoms, and no abdominal pain.

## 1. Introduction

Acute cholecystitis causes acute abdominal pain and may necessitate emergency surgery or intensive antibiotic therapy and percutaneous drainage, depending on the patient’s condition [1]. Early diagnosis of acute cholecystitis is needed to achieve good outcomes [2]. Typical symptoms of acute cholecystitis include acute-onset abdominal pain in the right upper quadrant and persistent fever [1,3]. Acute cholecystitis can be diagnosed through an effective physical examination and imaging techniques such as ultrasound and computed tomography [2,3,4]. Early diagnosis and treatment of acute cholecystitis improve patients’ quality of life and facilitate early discharge from the hospital [1]. However, older patients may show diverse and vague symptoms with an atypical clinical course [5,6]. The symptoms of cholecystitis are not clearly apparent in older patients, so its diagnosis can be difficult [7].

The symptoms of acute cholecystitis in older patients may be atypical and difficult to diagnose, leading to delayed treatment. The ability to perform activities of daily living decreases with age, and the number of bedridden patients increases [8,9]. In addition, a decline in cognitive function makes it difficult for older patients to convey their symptoms properly [10]. As older patients have multiple symptoms and visit primary care physicians and emergency departments, medical professionals can underestimate their symptoms [11,12]. Therefore, the diagnosis of acute diseases might be missed. Due to the vagueness of the pain caused by acute cholecystitis, fever may be the only symptom during the initial course of the disease [13]. Proper physical examination may lead to a diagnosis. However, the clinician must be trained in physical examination to correctly identify the symptoms indicative of acute cholecystitis in older patients.

Clarifying the relationship between the timing of acute cholecystitis diagnosis and patient factors among older patients could lead to early diagnosis and treatment. Acute cholecystitis can be made easier to appropriately suspect by enhancing the reliability and validity of the medical history and physical findings and being aware of patient factors related to the delay in diagnosis. As the population ages, many super-elderly people may be diagnosed with cholecystitis in rural community hospitals. However, there are no clear epidemiological data on the symptoms and timing of diagnosis. Clarifying the epidemiology and factors related to the timing of diagnosis of acute cholecystitis could reduce diagnostic errors. Therefore, this study aimed to explore the risk factors for delayed diagnosis of acute cholecystitis among older patients in a rural hospital.

## 2. Materials and Methods

This was a retrospective cohort study of patients aged over 65 years diagnosed with acute cholecystitis in a rural community hospital.

### 2.1. Setting

Unnan City is one of the most rural cities in Japan and is located in the southeast of Shimane Prefecture. In 2020, the total population of Unnan was 37,638 (18,145 men and 19,492 women), with 39% of the population aged over 65 years, which is expected to reach 50% by 2025. There are 16 clinics, 12 home care stations, 3 visiting nurse stations, and only 1 public hospital (Unnan City Hospital) [14]. At the time of this study, Unnan City Hospital had 281 beds, comprising 160 acute care beds, 43 comprehensive care beds, 30 rehabilitation beds, and 48 chronic care beds. There were 14 medical specialties, and the nurse-to-patient ratio was 1:10 for acute care, 1:13 for comprehensive care, 1:15 for rehabilitation, and 1:25 for chronic care.

### 2.2. Participants

All patients aged over 65 years old admitted to Unnan City Hospital between 1 April 2016, and 31 December 2021, were included in this study.

### 2.3. Measurements

#### 2.3.1. Primary Outcome

The primary outcome was the time from the initial symptoms to the diagnosis of acute cholecystitis. We reviewed the electronic medical records of patients with acute cholecystitis and investigated the timing of diagnosis, treatment, and symptoms onset. Cholecystitis diagnosis was confirmed based on the vital signs, abdominal physical findings (Murphy’s sign and right upper quadrant pain/tenderness), laboratory test results (white blood cell count), abdominal ultrasound, computed tomography findings according to the Tokyo criteria [4], and pathological findings for the gall bladder. Based on previous research, a duration of three days from symptom onset to diagnosis was related to mortality rate differences [15]. To assess the primary outcome, we divided the participants into two groups: the group with a time to diagnosis of >3 days (exposure) and the group with a time to diagnosis of ≤3 days (control) [15].

#### 2.3.2. Independent Variables

The following background data were collected from the electronic medical records at Unnan City Hospital: age; sex; body mass index (BMI); albumin level (g/dL) to assess the nutrition status [16]; serum creatinine level and estimated glomerular filtration rate to assess the renal function (mL/min/1.73 m^2^); and hemoglobin level (g/dL). Moreover, the care level was assessed based on the Japanese long-term insurance system [17], and the Charlson comorbidity index (CCI) score based on past medical history (history of heart failure, myocardial infarction, asthma, chronic obstructive pulmonary disease, kidney diseases, liver diseases, diabetes mellitus, brain infarction, brain hemorrhage, hemiplegia, connective tissue diseases, dementia, and cancer) was used to assess comorbidities severity [18]. The cognitive, motor, and total scores of the functional independence measure (FIM) at admission were measured by therapists as an indicator of the patient’s capacity to perform activities of daily living and their living situation (with/without family). The following clinical data were collected: the presence of abdominal pain, vital signs, time from initial presentation to definitive diagnosis, and the diagnosis decision. As variables related to the diagnosis of cholecystitis, data on the white blood cell count (×10^3^/μL) and total bilirubin (mg/dL), direct bilirubin (mg/dL), aspartate aminotransferase (IU/L), alanine aminotransferase (IU/L), and alkaline phosphatase (IU/L) levels were collected.

### 2.4. Analyses

The Student’s t-test was performed on parametric data, and the Mann–Whitney U test was performed on non-parametric data. Based on previous studies and the average values of the variables, numerical variables were dichotomized as follows: age (≥80 and <80 years), CCI (≥5 and <5) [17], and care level (≥1 and 0) [18]. A univariate regression model was used to assess whether the time to cholecystitis diagnosis was associated with the independent variables. Variables with statistically significant differences in the univariate regression analysis were further analyzed using a logistic regression model. We calculated the variance inflation factor (VIF) to investigate overfitting and multicollinearity. The VIF for the variables was less than 2; therefore, this analysis had a low risk of overfitting and multicollinearity. Patients with missing data were excluded from the analysis. Statistical significance was defined as a *p*-value < 0.05. All statistical analyses were performed using EZR (Saitama Medical Center, Jichi Medical University, Saitama, Japan), which is a graphical user interface for R (The R Foundation, Vienna, Austria) [19].

### 2.5. Ethical Considerations

The hospital was assured that the patients’ information would remain anonymous and confidential. Information related to this study was posted on the hospital website without disclosing any details about the patients. To address any questions regarding this study, the contact information of the hospital representative was listed on the website. All participants were informed of the purpose of this study, and informed consent was obtained from all participants. The clinical ethics committee of our institution approved this study (approval code: 20210026).

## 3. Results

### 3.1. Participant Demographics

Of the 15,296 patients admitted to the community hospital, 11,536 were older than 65 years. After excluding 11,401 patients with diagnoses other than cholecystitis, 135 were evaluated. The patient inclusion flowchart is presented in Figure 1.

The average ages of the control and exposure groups were 77.71 years (standard deviation [SD] = 14.62) and 80.13 years (SD = 13.95), respectively. The proportions of men in the control and exposure groups were 41.7% and 64.1%, respectively. Between the control and exposure groups, male sex, BMI, total FIM score, presence of abdominal pain, and serum albumin level showed statistically significant differences (Table 1).

### 3.2. Relationship between the Time to Diagnosis and Demographic Factors

The C-statistic for the logistic regression model was 0.747 (95% confidence interval [CI], 0.658–0.836). The logistic regression model revealed that the serum albumin level was significantly associated with a time to diagnosis > 3 days (odds ratio = 0.51; 95% Cl, 0.28–0.94). The other factors were not significantly related to a time to diagnosis > 3 days (Table 2).

## 4. Discussion

This study identifies potential risk factors for delayed diagnosis of cholecystitis among older patients in rural hospitals. The results show that a low serum albumin level might be related to delayed diagnosis, while male sex, the presence of abdominal pain, and high BMI showed trends toward early cholecystitis diagnosis. Clinicians should be concerned about delayed diagnosis of cholecystitis in older female patients with poor nutritional status, such as low serum albumin levels, and in patients with a low BMI and vague symptoms without abdominal pain.

Serum albumin levels could be affected by various metabolic factors, which might mask the symptoms of cholecystitis, except for abdominal pain. We found that high serum albumin levels are related to early diagnosis of cholecystitis among older admitted patients. Serum albumin levels may be related to nutritional status, with a low serum albumin level indicating a poor nutritional status [20,21]. A poor nutritional status can affect a person’s physical and psychological conditions [22,23]. Patients with poor nutritional status could show decreased sensitivity to external stimuli, causing decreased pain sensation and clarity of cognition [24,25,26]. Decreased pain sensation and poor cognition can make the symptoms of diseases vague and difficult to diagnose. Previous studies in rural areas have shown that low serum albumin levels are associated with high mortality and morbidity [20,21,27]. High mortality and morbidity rates might also be caused by misdiagnosis and mistreatment of patients [28]. When older patients with low serum albumin levels have vague symptoms, physicians should be cautious of delayed or missed diagnoses and investigate their symptoms comprehensively.

Male sex may affect the perception of systemic symptoms, which could affect the detection of cholecystitis. Male patients tend to experience less pain and are less sensitive to various symptoms than female patients [29]. As this article shows, men tend to have an early diagnosis of cholecystitis. In addition, older male patients tend to visit the hospital with more severe symptoms than female patients [30,31]. Medical professionals could be biased in their approach to analyzing symptoms and diagnosing diseases in older male patients [32]. In Japanese culture, masculinity may be dominant among the older generation; therefore, older male patients do not visit hospitals with mild symptoms [33,34]. The severity of the symptoms might explain why we observed a trend of early diagnosis in older male patients. Cultural issues and sex differences in symptom perception and help-seeking behaviors should be considered when diagnosing patients [35,36]. This study did not investigate cholecystitis severity directly; therefore, future studies should investigate the sex differences in the perceptions of symptoms of cholecystitis and other infectious diseases to adjust the bias of physicians and decrease the diagnostic delay.

Abdominal pain is one of the symptoms leading to a suspicion of cholecystitis, and the diagnosis can be confirmed through an effective physical examination. Precise physical examination is essential for effectively diagnosing acute cholecystitis [15]. In this study, the absence of subjective abdominal pain might be related to the delayed diagnosis, as the diagnosis was made based on objective physical findings of the abdomen. Various physical signs are observed in the right upper quadrant of the abdomen, such as Murphy’s sign, percussion tenderness, and liver knock pain [37]. Each physical finding had high specificity. Positive findings of these physical examinations increase the possibility of the cause of symptoms being acute cholecystitis. As this article shows, the lack of abdominal pain might be related to the delay in diagnosing acute cholecystitis. Physicians may not appropriately examine older patients without abdominal pain. Furthermore, they may not consider acute cholecystitis a differential diagnosis in patients without abdominal pain and fever [38]. To effectively diagnose acute cholecystitis on time, physicians should obtain clinical history and perform a physical examination in older patients with vague symptoms, considering the possibility of cholecystitis.

A high BMI could prompt physicians to examine patients intensively. This study showed a high BMI trend toward early diagnosis of acute cholecystitis among older patients. In this study, the average BMI was over 23, and the patients were slightly obese based on the Japanese standard BMI of 22. A high BMI may cause physicians to be more thorough than usual to avoid missing critical diseases [39,40]. Previous studies have shown associations between obesity and diagnostic errors by physicians, and physicians could be aware of this [39,40,41]. Therefore, physicians may examine relatively obese patients in depth to avoid diagnostic errors and order several tests, such as abdominal ultrasound and computed tomography, to check for abdominal inflammation [42,43]. In this study, physicians also investigated patients with high BMI more intensively than those with low BMI. Older patients with a high BMI who consider themselves healthy might have vague symptoms, be reluctant to seek medical help, and prefer self-management [44]. This could delay the detection of cholecystitis [15,37]. The patients’ BMI and appearance could affect physicians’ attitudes toward examining them; therefore, physicians should thoroughly examine all older patients with vague symptoms to avoid a delayed diagnosis of various critical diseases regardless of BMI [45]. The following studies investigated the relationship between diagnostic errors and BMI.

One limitation of this study is that it was performed at only one hospital in a rural Japanese setting. To elucidate the geographical variation in rural contexts, we collected data for a long period and included all admitted older patients with cholecystitis. Future studies should include more hospitals across prefectures and countries. As Japan is one of the leading countries in terms of population aging, our findings can be applied to countries preparing for aging-related issues. Furthermore, the retrospective cohort design of this study with a small sample size could not clearly reveal a cause-and-effect relationship. Future longitudinal studies should investigate this relationship.

## 5. Conclusions

A low serum albumin level may be a risk factor for delayed diagnosis of cholecystitis. Male sex, the presence of abdominal pain, and high BMI show trends toward early diagnosis. Clinicians should consider the possibility of cholecystitis in older female patients with poor nutritional status, such as low serum albumin levels, low BMI, and vague symptoms without abdominal pain.

## Figures and Tables

**Figure 1 medicina-58-01392-f001:**
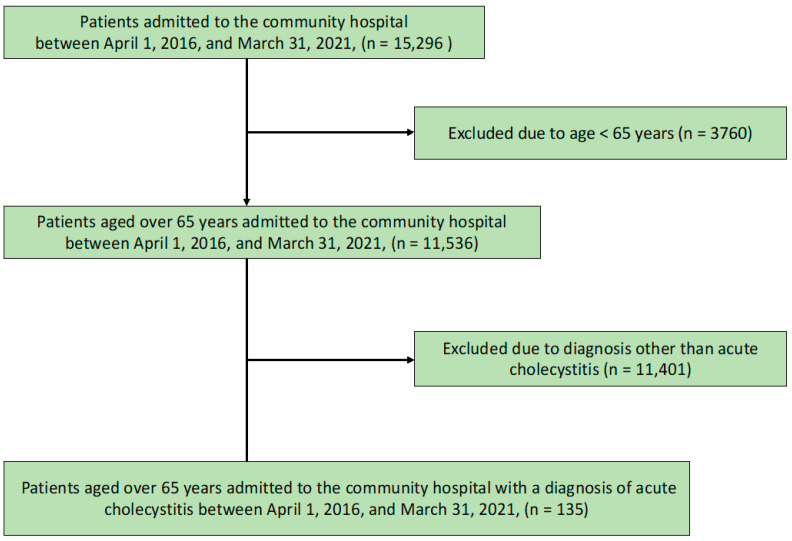
Flow chart showing the participant selection process.

**Table 1 medicina-58-01392-t001:** Participants’ demographics.

	Time to Diagnosis		
Factor	≤3 Days (Control)	>3 Days (Exposure)	*p*-Value
N	96	39	
Age, mean (SD)	77.71 (14.62)	80.13 (13.95)	0.379
Age ≥ 80 years (%)	51 (53.1)	24 (61.5)	0.446
Male sex (%)	40 (41.7)	25 (64.1)	0.023
Albumin level, mean (SD)	3.74 (0.67)	3.29 (0.73)	0.001
BMI, mean (SD)	23.68 (4.13)	21.70 (3.09)	0.008
Creatinine level (SD)	0.91 (0.48)	0.90 (0.92)	0.947
eGFR (SD)	64.48 (20.87)	64.16 (19.79)	0.935
Hemoglobin level, mean (SD)	12.96 (2.00)	12.39 (1.60)	0.119
White blood cell count (median)	10.15 (3.00, 29.70)	9.70 (3.60, 19.30)	0.638
Total bilirubin level (median)	1.30 (0.30, 7.00)	1.20 (0.30, 7.20)	0.602
Direct bilirubin level (median)	0.50 (0.10, 5.30)	0.60 (0.10, 4.40)	0.196
AST level (median)	34 (11, 2319)	56 (15, 1199)	0.187
ALT level (median)	34 (7, 849)	56 (8, 726)	0.437
ALP level (median)	284(53, 1719)	328 (70, 3232)	0.197
FIM score at discharge			
Total FIM score (median)	120 (18, 126)	80 (18, 126)	0.03
Motor domain score (median)	85 (11, 91)	63 (13, 91)	0.054
Cognitive domain score (median)	35.00 (5, 35)	32 (5, 35)	0.019
Abdominal pain (%)	70 (72.9)	19 (48.7)	0.009
Systolic blood pressure (SD)	130.23 (25.51)	126.92 (21.63)	0.478
Diastolic blood pressure (SD)	76.49 (14.70)	75.46 (16.34)	0.722
Heart rate (SD)	86.28 (18.65)	84.67 (17.85)	0.645
Fever (%)	32 (33.3)	14 (35.9)	0.842
Respiratory rate (SD)	20.07 (4.52)	20.44 (5.36)	0.76
SpO2 (SD)	96.09 (2.20)	95.51 (2.53)	0.194
CCI score ≥ 5 (%)	53 (55.2)	23 (59.0)	0.707
CCI score (%)			
1	7 (7.3)	2 (5.2)	
2	9 (9.4)	2 (5.1)	
3	6 (6.2)	4 (10.3)	
4	21 (21.9)	8 (20.5)	
5	14 (14.6)	8 (20.5)	
6	16 (16.7)	6 (15.4)	
7	10 (10.4)	4 (10.3)	
8	6 (6.2)	3 (7.7)	
9	2 (2.1)	2 (5.1)	
10	4 (4.2)	0 (0.0)	
11	1 (1.0)	0 (0.0)	
Heart failure (%)	13 (13.5)	7 (17.9)	0.594
MI (%)	6 (6.2)	1 (2.6)	0.673
Asthma (%)	5 (5.2)	2 (5.1)	1
Peptic ulcer (%)	19 (19.8)	8 (20.5)	1
Kidney disease (%)	13 (13.5)	2 (5.1)	0.23
Liver disease (%)	5 (5.2)	4 (10.3)	0.281
COPD (%)	3 (3.1)	1 (2.6)	1
DM (%)	19 (19.8)	7 (17.9)	1
Brain infarction (%)	22 (22.9)	4 (10.3)	0.147
Brain hemorrhage (%)	0 (0.0)	1 (2.6)	0.289
Hemiplegia (%)	8 (8.3)	3 (7.7)	1
Connective tissue disease (%)	2 (2.1)	1 (2.6)	1
Dementia (%)	21 (21.9)	11 (28.2)	0.504
Cancer (%)	17 (17.7)	3 (7.7)	0.184
Living with family	86 (89.6)	30 (76.9)	0.098
Dependent condition (%)	27 (28.1)	16 (41.0)	0.158
Care level (%)			
0	69 (71.9)	23 (59.0)	0.336
1	5 (5.2)	3 (7.7)	
2	7 (7.3)	6 (15.4)	
3	7 (7.3)	2 (5.1)	
4	6 (6.2)	2 (5.1)	
5	2 (2.1)	3 (7.7)	

Abbreviations: ALP, alkaline phosphatase; ALT, alanine aminotransferase; AST, aspartate aminotransferase; BMI, body mass index; CCI, Charlson comorbidity index; COPD, chronic obstructive pulmonary disease; DM, diabetes mellitus; eGFR, estimated glomerular filtration rate; FIM, functional independence measure; MI, myocardial infarction; SD, standard deviation; SpO2, peripheral oxygen saturation.

**Table 2 medicina-58-01392-t002:** Logistic regression model results.

Factor	Odds Ratio	95% CI	*p*-Value
Abdominal pain	0.57	0.24–1.39	0.22
Male sex	0.43	0.18–1.01	0.05
Albumin	0.51	0.28–0.94	0.032
BMI	0.90	0.80–1.01	0.07
Total FIM score	1.00	0.99–1.01	0.93

Abbreviation: CI, confidence interval.

## Data Availability

The datasets used and/or analyzed during the current study may be obtained from the corresponding author upon reasonable request.

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
