# Peer review of "Risk Factors for Delayed Diagnosis of Acute Cholecystitis among Rural Older Patients: A Retrospective Cohort Study"

_medicina, 2022, doi:10.3390/medicina58101392_

Round 1

Reviewer 1 Report (Previous Reviewer 2)

There are few language errors. I suggest authors to get it reviewed by a language expert.

Author Response

Responses to the reviewers’ comments

Thank you for reviewing our manuscript and providing suggestions for its improvement. We have provided point-by-point responses to the reviewers’ comments. Our revisions are highlighted in red font in this response letter and the manuscript. We hope the revised manuscript meets the journal’s requirements and can now be considered for publication.

There are few language errors. I suggest authors to get it reviewed by a language expert.

Response:

Thank you for your valuable feedback. Per your suggestion, professional English language editors have thoroughly reviewed and edited our manuscript.

Reviewer 2 Report (New Reviewer)

I read with interest the manuscript about risk factors for a delayed diagnosis of acute cholecystitis in elderly patients in rural area. However some major concerns must be addressed to be taken into consideration for publication:

- multivariate analysis of 5 variables and 39 events brings too high risk of overfitting of the model

- the authors didn't take into account the socio-economical and cultural background of patients and their family, than in rural areas is very important

-in the abstract and in the title, maybe could be more appropriated to talk about "risk factors" for delayed diagnosis, than "relationships"? Results must be more clear ("show a trend", what that "trend" scientifically mean?")

-The authors define a delayed diagnosis "based on previous research". Which previous research? please clarufy

-In the discussion authors say that this study "clarifyes" the relationships. Please, correct such strong conclusions, since according to the methods this study can't clarify any relationship (wider, perspective and more appropriated studies are needed)

-An extensive English editing is needed

_Why the authors didn't use Tokyo guidelines for diagnosis. Diagnostic criteria need to be more objective. There could be too much variability bringing an important bias

Author Response

Responses to the reviewers’ comments

Thank you for reviewing our manuscript and providing suggestions for its improvement. We have provided point-by-point responses to the reviewers’ comments. Our revisions are highlighted in red font in this response letter and the manuscript. We hope the revised manuscript meets the journal’s requirements and can now be considered for publication.

I read with interest the manuscript about risk factors for a delayed diagnosis of acute cholecystitis in elderly patients in rural area. However, some major concerns must be addressed to be taken into consideration for publication:

-Multivariate analysis of 5 variables and 39 events brings too high risk of overfitting of the model

Response:

Thank you for pointing this out. We have repeated the analysis and calculated the variance inflation factor (VIF) to investigate overfitting and multicollinearity. The VIF was less than 2; therefore, we considered that our analysis had a low risk of overfitting. The C-statistic was also calculated. We have added this information to the manuscript as follows:

“We calculated the variance inflation factor (VIF) to investigate overfitting and multicollinearity. The VIF for the variables was less than 2; therefore, this analysis had a low risk of overfitting and multicollinearity.” (Lines 121 to 124)

“The C-statistic for the logistic regression model was 0.747 (95% confidence interval [CI], 0.658–0.836).” (Lines 158 to 159)

-The authors didn't take into account the socio-economical and cultural background of patients and their family, than in rural areas is very important.

Response:

Thank you for your insightful comment. We have added the variable of the patients’ living situation (with/without family).

“The following background data were collected from the electronic medical records at Unnan City Hospital: age; sex; body mass index (BMI); albumin level (g/dL) to assess the nutrition status [16]; serum creatinine level and estimated glomerular filtration rate to assess the renal function (mL/min/1.73 m2), and hemoglobin level (g/dL). Moreover, the care level was assessed based on the Japanese long-term insurance system [17], and the Charlson comorbidity index (CCI) score based on the past medical history (history of heart failure, myocardial infarction, asthma, chronic obstructive pulmonary disease, kidney diseases, liver diseases, diabetes mellitus, brain infarction, brain hemorrhage, hemiplegia, connective tissue diseases, dementia, and cancer) was used to assess comorbidities severity [18]. The cognitive, motor, and total scores of the functional independence measure (FIM) at admission were measured by therapists as an indicator of the patient’s capacity to perform activities of daily living and their living situation (with/without family).” (Lines 97 to 108)

-In the abstract and in the title, maybe could be more appropriated to talk about "risk factors" for delayed diagnosis, than "relationships"? Results must be clearer ("show a trend", what that "trend" scientifically mean?")

Response:

Thank you for your suggestion. We have revised the title and abstract according to your suggestions as follows:

“Clarifying the risk factors for delayed diagnosis among older patients could lead to early diagnosis and treatment of acute cholecystitis. This study aimed to explore the risk factors for delayed acute cholecystitis diagnosis among rural older patients.” (Lines 15 to 17)

“Low serum albumin levels are related to delayed cholecystitis diagnosis and male sex. The presence of abdominal pain and a high body mass index (BMI) may be related to early cholecystitis diagnosis. Clinicians should be concerned about the delay in cholecystitis diagnosis in older female patients with poor nutritional conditions, including low serum albumin levels, a low BMI, vague symptoms, and no abdominal pain.” (Lines 26 to 30)

-The authors define a delayed diagnosis "based on previous research". Which previous research? please clarify

Response:

Thank you for the valuable feedback. To clarify this point, we have added the following information to the manuscript regarding the cholecystitis diagnosis method:

“Cholecystitis diagnosis was confirmed based on the vital signs, abdominal physical findings (Murphy’s sign and right upper quadrant pain/tenderness), laboratory test results (white blood cell count), abdominal ultrasound, computed tomography findings according to the Tokyo criteria [4], and pathological findings of the gall bladder. Based on previous research, a duration of three days from symptom onset to diagnosis was related to mortality rate differences [15].” (Lines 88 to 93)

-In the discussion authors say that this study "clarifies" the relationships. Please, correct such strong conclusions, since according to the methods this study can't clarify any relationship (wider, perspective and more appropriated studies are needed)

Response:

Thank you for pointing this out. Per your comment, we have revised the discussion as follows to ensure modest inferences and submissions:

“A low serum albumin level may be a risk factor for delayed cholecystitis diagnosis. Male sex, the presence of abdominal pain, and high BMI show trends toward early diagnosis. Clinicians should consider the possibility of cholecystitis in older female patients with poor nutritional status, such as low serum albumin levels, low BMI, and vague symptoms without abdominal pain.” (Lines 243 to 247)

-An extensive English editing is needed

Response:

Thank you for your recommendation. Per your suggestion, our manuscript has been thoroughly reviewed and edited by professional English language editors.

-Why the authors didn't use Tokyo guidelines for diagnosis. Diagnostic criteria need to be more objective. There could be too much variability bringing an important bias

Response:

Thank you for the insightful comment. To rectify this, we have reviewed the medical records data and added a description of the Tokyo criteria for cholecystitis diagnosis as follows:

“Cholecystitis diagnosis was confirmed based on the vital signs, abdominal physical findings (Murphy’s sign and right upper quadrant pain/tenderness), laboratory test results (white blood cell count), abdominal ultrasound, computed tomography findings according to the Tokyo criteria [4], and pathological findings of the gall bladder. Based on previous research, a duration of three days from symptom onset to diagnosis was related to mortality rate differences [15].” (Lines 88 to 93)

Round 2

Reviewer 2 Report (New Reviewer)

I think the manuscript has improved. All requests were addressed.

Please, correct the sentences where you wrognly talk about "...delayed cholecystitis..." instead of "delayed DIAGNOSIS of acute cholecystitis, it is very different (such as in line 6 in the abstract "...the  risk factors for delayed acute cholecystitis diagnosis among rural older patients...").

Author Response

Responses to the reviewer’s comments

Thank you for reviewing our manuscript and providing suggestions for its improvement. We have provided point-by-point responses to the reviewer’s comments. We hope the revised manuscript meets the journal’s requirements and can now be considered for publication.

I think the manuscript has improved. All requests were addressed.

Please, correct the sentences where you wrongly talk about "...delayed cholecystitis..." instead of "delayed DIAGNOSIS of acute cholecystitis, it is very different (such as in line 6 in the abstract "...the risk factors for delayed acute cholecystitis diagnosis among rural older patients...").

Response:

Thank you for your valuable feedback. Per your suggestion, we have revised all of the parts of delayed cholecystitis diagnosis to delayed diagnosis of cholecystitis.

This manuscript is a resubmission of an earlier submission. The following is a list of the peer review reports and author responses from that submission.

Round 1

Reviewer 1 Report

I thank the authors for the opportunity to evaluate their work.
Due to the progressive aging of the world population, the diagnosis and treatment of acute cholecystitis in the elderly is an increasingly relevant area of study.

Unfortunately, I believe that this paper has major serious structural problems.

The study assumes that in the elderly (defined as all those older than 65 years) because the symptoms of cholecystitis may be atypical the clinical evaluation may not lead to a correct diagnosis. It also completely overlooks the role that the patient's medical history and blood test results can play.
The stated aim of the study is as follows: : “Therefore, this study aimed to explore the relationship between the symptoms and demographic data of acute cholecystitis among older patients in rural community hospitals and the timing of the diagnosis from the onset of symptoms. “
The development of the study does not seem to meet the stated aim.
The study attempts to validate a cause-and-effect relationship between altered albumin value and pain stimulus response ( the unit of measurement of serum albumin is not specified); concluding that a low albumin level might be associated with a delayed diagnosis of cholecystitis and that patients with a low BMI are evaluated less accurately than those with a high BMI.These conclusions were obtained from analysis of data from a single-center retrospective study of 135 patients recruited over 68 months.
I believe that the study design is not adequate to aim to achieve its objective.
Acute cholecystitis is a condition that is the subject of numerous papers in the literature and for which multiple diagnostic flowcharts have been established (e.g., Tokyo Guidelines 2018).
The patient's history, values of white blood cells, liver function tests, CRP, and ultrasonography succeed in the vast majority of cases in leading to a correct diagnosis even in patients over 80 years of age and atypical symptoms.
The statements about the clinical examination and the patient's BMI seem to be intended to support the professional inadequacy of those performing it

Clinical suspicion in differential diagnosis accompanied by proper application of a diagnostic flowchart is certainly more effective than albumin testing in these patients.

Reviewer 2 Report

There are few language and grammatical errors. See line 24 in Abstract, 65-66 in Introduction and lines 162-163 in Discussion as some examples. 

Abstract

1.     The statement that acute cholecystitis requires emergency surgery is misleading. There is a definite role of a non-operative treatment in acute setting and delayed cholecystectomy in electively. Please correct the statement accordingly in 'Abstract' and 'Introduction'.

Introduction

1.     The study was conducted in single hospital but the line 69 suggests that it is a multicenter study, please rectify.

Material and methods

1.     Expand ADL (line 111). 

Discussion

1.     The finding of association of low serum albumin with delayed diagnosis and high BMI with early diagnosis may be just chance findings as the sample size is small. This may not be applicable to a larger similar population. The p values of BMI and male gender (Table 2) also do not suggest association of these factors with duration to make diagnosis. 

2.     The statement regarding poor nutritional status and clarity of consciousness requires more deliberation (lines 168-169). A person with poor nutrition may not necessarily have altered consciousness. 

3.     Line 176-177 is incomprehensible and requires paraphrasing. 

4.     The third paragraph in this section is unclear and requires rephrasing. On one hand, the study describes that male gender is associated with an early diagnosis of cholecystitis yet the description in this paragraph suggest the opposite.  

5.     How the patients without abdominal pain were diagnosed? This subgroup of patients must be separately analyzed to identify the factors responsible for delayed diagnosis, like altered consciousness, sepsis or other cognitive or neural disorders. 

6.     The fourth and fifth paragraphs on association between BMI and early diagnosis are also vague. It has been quoted that obesity can lead to missed diagnosis, yet higher BMI (which again is not statistically significant in the reported cohort) has been reported to be associated with an early diagnosis. This may also convey a wrong message that the person with low BMI may not require a diligent examination. 

References

1.     Some of the references are too old to be included. Consider replacing reference numbers 1,2,13,18,21 and 40 with newer ones unless they are of historical importance.